# The Effect of Changes in Employment on Health of Work-Related Injured Workers: A Longitudinal Perspectives

**DOI:** 10.3390/healthcare9040470

**Published:** 2021-04-15

**Authors:** Han-Kyoul Kim, Kyu-Min Kim, Jae-Hak Kim, Hyun-Sill Rhee

**Affiliations:** 1Department of Rehabilitation Medicine, Seoul National University Hospital, Seoul 03080, Korea; collkhk@gmail.com; 2National Traffic Injury Rehabilitation Research Institute, National Traffic Injury Rehabilitation Hospital, Yang-Pyeong 12564, Korea; 3Department of Health Policy and Management, College of Health Science, Korea University, Seoul 02841, Korea; perves@korea.ac.kr (K.-M.K.); rlawogkr12@korea.kr (J.-H.K.); 4Fitness & Rehabilitation Exercise Department in National Rehabilitation Center, Seoul 01022, Korea; 5School of Health Policy and Management, College of Public Health Science, Korea University, Seoul 02841, Korea

**Keywords:** work-related injuries, occupational injuries, employment status, health status, propensity score matching

## Abstract

This longitudinal study attempted to identify changes in employment status and overall health status. The participants were workers who had experienced work-related injuries in the past. In this study, we used the Panel Study of Workers’ Compensation Insurance from 2013 to 2017. This study utilized propensity score matching for a quasi-experimental design study of the first year to exclude the effects of the confounding variables and exclude the effect of employment status, which is the main independent variable. After applying propensity score matching the research subjects totaled 1070. Changes in employment status were found to have a negative effect on overall health status. This raises new implications for existing industrial accident-related support policies. Thus, it is considered that the scope should be expanded from policies related to re-employment of workers after an industrial accident to improving quality of life through maintaining employment from a long-term perspective. The notable point of this study was to apply the PSM methods. By applying PSM, we clearly identified the effect of changes in employment status on health status.

## 1. Introduction

In society, labor is an important factor in forming an individual’s life in relation to economic and social activities. The factors related to labor include employment type, wage level, and working environment; these factors are reported to affect the health of individuals directly or indirectly [1,2,3]. Instability of employment such as precarious or irregular employment is negatively related to an individual’s mental and physical health [4,5]. Many studies confirmed that employment status is a risk factor for health outcomes [6,7,8]. A study in Sweden, analyzing the association between precarious employment of young adults aged 18–34 years and its effects on their mental health, showed that the incident rate ratio (IRR) was 1.4 times higher than in the stable employment group [7]. In addition, precarious employment had a negative impact on health status after adjusting previous health conditions and was found to be related to psychosocial disorders and reported to have a negative association with subjective health status [8,9]. Workers who are employed precariously have shown negative health outcomes such as physical function, depression and suicidal ideation [10]. In particular, employment is highly correlated with health changes and has recently become an important factor in Korean society [11].

According to a survey by the Ministry of Employment and Labor in Korea, the number of workers who experienced industrial accidents—either due to an illness or by accident—increased from 89,848 in 2017 to 102,305 in 2018, and the estimated economic loss also increased by KRW 22.2 trillion in 2017 and 25.2 trillion won in 2018. Even after considering the expansion of industrial accident insurance policies, the number of work-related injuries is high and increasing steadily each year in Korea. In particular, it is known that people who experience injuries at work exhibit psychologically unstable characteristics compared to healthy people and thus have a high possibility of it manifesting into more serious forms, such as depression [12]. Psychological factors are known to be factors that prevent them from returning to work and getting a new job [13]. In addition, workers who experience industrial accidents become socially isolated after the accident in correlation with anxiety, compulsion, hostility, stress, and depression [14].

In order to reduce the problems caused by industrial accidents, groundwork such as the strengthening of professional rehabilitation treatment, support for the return to work, and vocational training were provided based on the 5th Industrial Accident Insurance Rehabilitation Plan: 2018–2022 so that workers who experienced industrial accidents could return to work and society as full-fledged members. However, the industrial accident workers’ employment return rate was 61.9% in 2016, return to the original workplace was 41.4%, and about 40% of industrial accident workers were unable to return to work [15]. Despite such diverse efforts, more than half of them fail to return and experience difficulties returning to the original workplace. This can also be confirmed in a study using the Panel Study of Workers’ Compensation Insurance (PSWCI) data. Based on return-to-work situations, 277 out of 695 workers (39.9%) returned to their original workplace, and 418 out of 642 workers (65.1%) were employed again [16].

Based on the previous studies, it was found that precarious employment and health are closely related. While various short-term, small-sample, and cross-sectional studies on the association employment and health have been conducted in South Korea, there is a lack of clarity on the people who experienced industrial accidents and nationwide longitudinal association. This study aimed to identify the association between the changes in employment and health status in injured workers.

## 2. Materials and Methods

### 2.1. Data Source and Study Population

This study was a longitudinal study using the first to fifth years of the Panel Study of Workers’ Compensation Insurance (PSWCI) data on workers who had experienced work-related injuries. Because of the purpose of this study, identifying the association between changes in employment status and health, we had to use longitudinal data of less than three years. The PSWCI notes the impact of industrial accidents on individual workers, their families, society, and the state caused by changes in industrial and labor market structures and social environments today. It provides basic data that include a wide range of socioeconomic characteristics of industrial workers and their return to work, which can be used to establish policies related to employment, labor, and industrial accidents. The 1958 participants in 2012 were workers who experienced work-related injuries before applying propensity score matching. In 2012, the participants consisted of 1387 employed workers and 571 unemployed workers. After we applied the PSM method, the final participants in this study were 571 employed workers and 571 unemployed workers.

### 2.2. Variable Description

Overall health status was used as an outcome variable which is measured on a four-point scale for a single question, “How is your overall health status?” These single questions showed close correlation with objective health status in prior study [17]. Thus, the overall health status variables investigated in a single question can also be used as variables for health outcomes. As an independent variable, we used the employment status of injured workers. Although the employment status includes various types—employed vs. unemployed, regular employment vs. irregular employment—we introduced employment or unemployment as the independent variable. Based on the identification code of the first year’s data applied with propensity score matching, the data from the second to fifth survey were merged, and the form of the completed data was composed of unbalanced data.

### 2.3. Statistical Analysis

This study performed propensity score matching (PSM) for accurate comparison between groups after industrial accidents. By applying propensity score matching, a type of quasi-experimental design, it was assumed that the characteristics of the two groups in this study were the same—demographic, socioeconomic, and health behavior—except for whether they were employed or not. Based on the identification code of the first year’s data applied with propensity score matching, the data from the second, third, fourth, and fifth years were combined, and the form of the completed data was composed of unbalanced data. The study used nearest neighbor matching with a caliper (1:1), a matching method that can reduce the occurrence of convenience, focusing on minimizing heterogeneity between groups according to employment status. To identify changes in employment status and general health status of workers with industrial accidents, the study used the STATA 13.0 program (Stata Corp SE, College Station, TX, USA) to perform the following analysis. First, an annual descriptive statistics analysis was conducted to identify the characteristics of the research targets. Next, *t*-test and one-way ANOVA were identified as the characteristics of the overall health status according to the characteristics of the research subjects. Finally, a panel regression analysis was performed to confirm changes in employment status and general health status. A random-effect model was applied to the panel regression analysis.

## 3. Results

### 3.1. General Characteristics of Workers with Industrial Accident Experience

The general characteristics of each year of research subjects are described in Table 1. The overall health status of workers who experienced industrial accidents was 2.5 to 2.6 points lower than the average. In the case of employment status, the number of work hours increased from 70.8% in 2013 to 81% as time passed from the accident. In terms of personal characteristics, the proportion of men was around 84%. The distribution of education levels showed that the highest number of workers were high school graduates or higher, followed by elementary, middle school, and college graduates. In terms of the distribution of chronic diseases, about 83% of the panel subjects did not have chronic diseases in 2013, but of those, only 62.2% did not have chronic diseases in 2017. In other words, it seems that illnesses have been developing since the industrial accident and progressing into complex, chronic diseases. Furthermore, it was found that 65.5% of workers experienced work-related injuries within one year of employment. Similarly, in the following years, about 65% experienced accidents within one year of employment. Regarding accident type, industrial accidents caused by accidents accounted for an absolute majority with about 91% in all years, and industrial accidents caused by diseases were reported to be less than 10% in all years. About 57% of workers who experienced industrial accidents received treatment covered by industrial insurance, and 31% of those covered were treated for less than six months to one year, whereas only about 10% of the total were treated for more than a year. Finally, most of the subjects were found to have obtained disability ratings after experiencing industrial accidents (approximately 82%).

### 3.2. Overall Health Status Based on the General Characteristics of the Research Subjects

The overall health status scores, based on the characteristics of the research subjects as of 2017, are described in Table 2. First of all, the overall health score of the group employed after the industrial accident was higher, showing a significant difference from that of the unemployed (t = 16.185, *p* < 0.001). In terms of gender, men scored slightly higher than women (t = 2.025, *p* < 0.043). We could see that the health score dropped significantly as age increased (f = 51.82, *p* < 0.001). For education levels, the lowest was 2.3(0.7) points for elementary and lower graduates, and 2.9(0.7) points for college and higher groups (f = 57.16, *p* < 0.001). In other words, scores of overall health status were reported to be higher as education levels increased. Regarding overall health status, according to the number of chronic diseases, groups without chronic diseases scored higher; the larger the number of chronic diseases, the lower the score (F = 123.49, *p* < 0.001). In the distribution, according to the working period before industrial accidents, the score of the group that suffered industrial accidents was lower for those employed more than a year than when the work was less than one year (t = −4.84, *p* < 0.001). Meanwhile, for industrial accident type, the score of workers who experienced industrial accidents due to diseases was 2.5(0.7), which was lower than for those due to accidents (t = 2.82, *p* < 0.01). Regarding the difference between post-industrial care periods, groups with relatively short care periods of six months or less scored the highest (m = 2.8, s.d. = 0.6), and those with a period of care exceeding one year scored the lowest (m = 2.2, s.d. = 0.8). This showed a significant statistical difference (F = 47.68, *p* < 0.001). Finally, the scores of groups with disability ratings were lower than those without (t = −5.74, *p* < 0.001).

### 3.3. A Longitudinal Association Analysis between Precarious Employment and Overall Health Status

The results of the change in employment status and overall health status are described here. The final analysis was conducted with a hierarchical regression model to identify the impact of changes in employment status. Model 1 is a result of identifying the effect of employment status on health status when input is independent, and it was shown that changes in employment status have a negative effect on general health status. In other words, the health status of the unemployed group was worse than that of the employed group. The results of Model 2, which put socioeconomic status variables as controls, also showed that employment status variables were significantly worse in terms of health for the unemployed group than for the employed group. Model 3 shows that even after inputting variables that reflect the characteristics of industrial accident workers, change in employment status resulted in the same negative change in the health status of the unemployed group. Finally, Model 4 was shown to have a negative effect on overall health status as a result of inputting health status variables. Looking at the results of the control variables, for socioeconomic conditions, the higher the age, the more negative the overall health status. In addition, higher education levels have a positive effect on health status. Variables related to industrial accidents showed that the health status of groups with more than one year of work before industrial accidents is better compared to those with less than one year of work. That is, the shorter the time employed prior to an industrial accident, the more negative the health status outcome. On the other hand, it was found that industrial accidents caused by diseases have a greater negative effect on health status compared to those caused by an accident. The group requiring a care period of six months or more and the group with a disability rating showed poor health status. When it comes to drinking, drinking groups have better health status than non-drinking groups.

## 4. Discussion

This study was a longitudinal study to identify the association between the employment status and overall health status of workers who experienced work-related injuries in Korea. In addition, we applied propensity score matching to balance employment status based on the starting point of the research data. Propensity score matching, a quasi-experimental study design that minimizes the effects of other confounding variables except for major treatments, is a suitable methodology for causality studies. The following are policy suggestions based on the results of this study.

First, it was found that employment change, the main purpose of this study, and overall health status were closely related. From 2013 to 2017, the average overall health status score was about 2.6, which did not change significantly (Table 1). However, when we analyzed the employment status and overall health status, it was found that the more unemployed the group, the worse the health status (Table 2). In addition, as a result of longitudinally confirming the effect of employment status on overall health status, it was found that unemployment worsened overall health status. The association between employment status and overall health status is consistent with the results of previous studies [18,19,20]. Studies on employment status, health, and quality of life of people with multiple neurological disorders show that those with certain disabilities have more experience with job loss, which results in deterioration of health status and deterioration in quality of life [18]. However, some studies on the employment of people with neurological disorders argued that many people with disabilities were still employed, and additional political supports such as vocational rehabilitation were needed to maintain and improve their employment [21,22]. According to a study on industrial workers’ return to work, it was reported that workers who have experienced injuries at the workplace are in a vicious cycle of unemployment and deterioration of health status when they are out of the period of receiving industrial accident compensation insurance [23]. In other words, if workers who have experienced industrial accidents are covered by industrial accident insurance, they can achieve personal and social milestones, such as returning to work. However, if insurance is excluded due to the termination of the coverage term for industrial accident insurance, that seems to put workers at risk for unemployment. This suggests that the scope of industrial accident insurance should be extended not only to workers’ return to work but also to maintaining employment status. From 2013 to 2017, the proportion of people in employment increased compared to those in unemployment. However, additional analysis of the 2013–2014 data shows that change from employment to unemployment is 4.7%. Therefore, to achieve employment maintenance of those employed, environmental factors such as relationships with supervisors and working conditions have to be considered [24].

Secondly, it is necessary to pay attention to the period of work before experiencing industrial accidents among the results of this study. About 65% of workers who experienced industrial accidents were injured within six months of starting work. This supports the fact that unskilled workers are more prone to injury than skilled workers. In addition, it seems that work is performed in a state where there is a lack of adequate education and training for the work. On the other hand, the overall health status of the group experiencing industrial accidents in the unskilled state was significantly worse (Table 2). The results of the panel regression model also show that overall health was worse as the workers experienced industrial accidents in the unskilled state (Table 3). Workers with a shorter work period are a relatively high-risk group [25]. In addition, according to a study on the types of employment of unskilled workers, it is clear that the social atmosphere in which the demand for skilled workers increases, during the economic crisis, relatively worsens the employment situation of unskilled workers [26]. Considering this social atmosphere, it is thought that entry barriers to re-employment will be high for unskilled workers who have experienced industrial accidents even after receiving the benefits of industrial accident insurance. Indeed, the employment rate of injured workers with a working period of more than six months was 79.2%, whereas for those with less than six months was 64.1%. We present the need for follow-up studies of workers who have experienced industrial accidents in the unskilled state. At the same time, this result demonstrates the need for differentiated social services and employment support services at the workers’ compensation level for unskilled workers.

Thirdly, we identified changes in health status according to the type of industrial accident. Based on the results of the first year’s analysis of this study, 1792 workers (91.5%) suffered industrial accidents due to accidents, and 166 workers (8.5%) suffered from diseases (Table 1). In addition, even after PSM was applied, the disease among the causes of industrial accidents has a negative effect on health in Korea. According to the Ministry of Employment and Labor’s Industrial Accident Analysis Report [27], the number of occupational diseases in Korea was 24.9%. However, the standards for industrial accidents caused by diseases are very demanding, and the accreditation and verification process is very complicated [28]. In the case of industrial accidents caused by diseases, the number can be underestimated compared to the actual number of patients due to the difficulty of verification and recognition [29]. Many studies about systematic risk underestimation of worker’s risk suggested that people make systematic errors in their perception and predictions as current and past emotions influence assessments. In other words, the frequency of dramatic or sensational events such as causes of death was more overestimated, and the frequency of less well-publicized causes such as stroke and asthma was more underestimated [30,31,32]. The results of this study show that the health status of groups who experienced industrial accidents caused by diseases was worse than those who experienced industrial accidents caused by accidents. That is, we identified that even in the case of industrial accidents, health status varies depending on the type of industrial accident. These results indicate that in-depth investigation and research are needed for the group of industrial workers that have experienced industrial accidents due to disease.

Meanwhile, this study has some limitations. Because secondary data (the panel study of workers’ compensation insurance) were used, it was not possible to analyze the factors influencing workers’ health status extensively. In other words, there were limits to the available variables. For example, there was a lack of variables that could reflect material status (e.g., household income), family, and social networks. The employment group was divided into restricted groups, and it was limited to reflecting all employment instability. In addition, due to the sporadic nature of the data, it was not suitable to confirm the continuous impact of employment instability.

## 5. Conclusions

The central contributions of this study are as follows. The advantage of this study is that, unlike previous studies that focused on returning to work after an industrial accident, changes in employment status were also considered. The results of the study suggest that longitudinal and consecutive political support and subsidy will be required to workers who experienced work-related injuries, even if they returned to the work. Lastly, the necessity for future research tasks was raised by referring to the state of health and employment in terms of industrial accident type and working period. Based on this, it is necessary to conduct research on post-insurance benefits for workers with industrial accidents and, at the same time, additional research on individual trait groups.

## Figures and Tables

**Table 1 healthcare-09-00470-t001:** General characteristics of workers with industrial accident experience.

	Year	2013 (*n*, %)	2014 (*n*, %)	2015 (*n*, %)	2016 (*n*, %)	2017 (*n*, %)
Variable						
General health status		2.5(0.7)	2.6(0.7)	2.6(0.7)	2.6(0.7)	2.6(0.7)
Economic status	Employment	1387(70.8)	1445(80.1)	1371(80.5)	1355(81.6)	1310(81.1)
	Unemployment	571(29.2)	358(19.9)	333(19.5)	305(18.4)	306(18.9)
Socioeconomic status						
Sex	Male	1649(84.2)	1514(84.0)	1422(83.5)	1382(83.3)	1337(82.7)
	Female	309(15.8)	289(16.0)	282(16.5)	278(16.7)	279(17.3)
Age	Under 30	108(5.5)	78(4.3)	59(3.5)	46(2.8)	30(1.9)
	30~39	288(14.7)	242(13.4)	200(11.7)	174(10.5)	162(10.0)
	40~49	513(26.2)	443(24.6)	407(23.9)	370(22.3)	337(20.9)
	50~59	697(35.6)	692(38.4)	597(35.0)	574(34.6)	533(33.0)
	Over 60	352(18.0)	348(19.9)	441(25.9)	496(29.9)	554(34.3)
Education	Under elementary	402(20.5)	369(20.5)	354(20.8)	348(21.0)	345(21.3)
	Middle school	373(19.1)	345(19.1)	324(19.0)	316(19.0)	305(18.9)
	High school	886(45.3)	814(45.1)	758(44.5)	738(44.5)	715(44.2)
	Over college	297(15.2)	275(15.3)	268(15.7)	258(15.5)	251(15.5)
Injury-related variables						
Working period	Under 1 year	1283(65.5)	1172(65.0)	1110(65.1)	1071(64.5)	1029(63.7)
	Over 1 year	675(34.5)	631(35.0)	594(34.9)	589(35.5)	587(36.3)
Cause of injury	Accident	1792(91.5)	1645(91.2)	1564(91.8)	1524(91.8)	1479(91.5)
	Disease	166(8.5)	158(8.8)	140(8.20	136(8.2)	137(8.5)
Claim duration	Under 6 months	1127(57.6)	1032(57.2)	986(57.9)	960(57.8)	932(57.7)
	Under 1 year	629(32.1)	579(32.1)	543(31.9)	529(31.9)	519(32.1)
	Over 1 year	202(10.3)	192(10.6)	157(10.3)	171(10.3)	165(10.2)
Health status						
Disability rating	Yes	1616(82.5)	1491(82.7)	1401(82.2)	1365(82.2)	1330(82.3)
	No	342(17.5)	312(17.3)	303(17.8)	295(17.8)	286(17.7)
Chronic disease	None	1630(83.2)	1413(78.4)	1216(71.4)	1116(67.2)	1005(62.2)
	1	252(12.9)	290(16.1)	344(20.2)	378(22.8)	402(24.9)
	2	62(3.2)	80(4.4)	98(5.80)	114(6.9)	134(8.3)
	Over 3	14(0.7)	20(1.1)	46(2.70)	52(3.1)	75(4.6)
Smoking	No	1006(51.4)	994(55.1)	1035(60.7)	1033(62.2)	1058(65.5)
	Yes	952(48.6)	809(44.9)	669(39.3)	627(37.8)	558(34.5)
Drinking	No	544(27.8)	595(33.0)	549(32.2)	618(37.2)	563(34.8)
	Yes	1414(72.2)	1208(67.0)	1155(67.8)	1042(62.8)	1053(65.2)

**Table 2 healthcare-09-00470-t002:** Overall health status based on the general characteristics of the research subjects.

	Year	2013		2014		2015		2016		2017	
Variable											
Economic status	Employment	2.7(0.6)	T = 15.89 ***	2.7(0.6)	T = −13.78 ***	2.7(0.6)	T = −15.38 ***	2.8(0.6)	T = −15.98 ***	2.8(0.6)	T = −16.19 ***
	Unemployment	2.1(0.7)		2.2(0.8)		2.1(0.8)		2.0(0.8)		2.0(0.8)	
Socioeconomic status											
Sex	Male	2.5(0.7)	T = −0.29	2.6(0.7)	T = 2.20 *	2.6(0.7)	T = 2.07 *	2.6(0.7)	T = 2.35 *	2.7(0.7)	T = 2.03 *
	Female	2.5(0.7)		2.5(0.7)		2.5(0.6)		2.5(0.7)		2.6(0.7)	
Age	Under 30	2.8(0.6)	F = 25.32 ***	2.9(0.7)	F = 38.40 ***	3.1(0.6)	F = 38.89 ***	3.2(0.5)	F = 44.24 ***	3.2(0.6)	F = 51.82
	30~39	2.8(0.6)		2.9(0.6)		2.9(0.6)		2.9(0.6)		3.0(0.6)	
	40~49	2.5(0.6)		2.7(0.6)		2.7(0.6)		2.8(0.6)		2.9(0.6)	
	50~59	2.5(0.6)		2.5(0.7)		2.6(0.7)		2.6(0.7)		2.7(0.6)	
	Over 60	2.3(0.6)		2.4(0.7)		2.3(0.7)		2.4(0.7)		2.4(0.7)	
Education level	Under elementary school	2.3(0.7)	F = 27.01 ***	2.3(0.7)	F = 58.82 ***	2.3(0.7)	F = 47.41 ***	2.3(0.7)	F = 49.76 ***	2.3(0.7)	F = 57.16 ***
	Middle school	2.5(0.7)		2.5(0.6)		2.5(0.7)		2.5(0.6)		2.5(0.7)	
	High school	2.6(0.7)		2.7(0.6)		2.7(0.7)		2.7(0.7)		2.8(0.6)	
	Over college	2.7(0.6)		2.9(0.7)		2.8(0.6)		2.8(0.6)		2.9(0.7)	
Injury-related variables											
Working period	Under 1 year	2.5(0.7)	T = 5.37 ***	2.6(0.7)	T = −5.24 ***	2.5(0.7)	T = −5.76 ***	2.6(0.7)	T = −5.66 ***	2.6(0.7)	T = −4.84 ***
	Over 1 year	2.6(0.6)		2.7(0.6)		2.7(0.6)		2.8(0.7)		2.8(0.6)	
Cause of injury	Accident	2.5(0.7)	T = 3.64 ***	2.6(0.7)	T = 1.91	2.6(0.7)	T = 2.23 *	2.6(0.7)	T = 2.62 **	2.7(0.7)	T = 2.82 **
	Disease	2.3(0.7)		2.5(0.7)		2.5(0.7)		2.5(0.7)		2.5(0.7)	
Claim duration	Under 6 months	2.7(0.6)	F = 93.13 ***	2.7(0.6)	F = 61.52 ***	2.7(0.6)	F = 55.09 ***	2.7(0.6)	F = 49.64 ***	2.8(0.6)	F = 47.68 ***
	6 months~year	2.4(0.6)		2.6(0.7)		2.5(0.7)		2.6(0.7)		2.6(0.7)	
	Over 1 year	2.0(0.7)		2.2(0.8)		2.2(0.8)		2.2(0.8)		2.2(0.8)	
Health status											
Disability	No	2.8(0.7)	T = 7.27 ***	2.8 (0.7)	T = −4.48 ***	2.8(0.7)	T = 5.32 ***	2.8 (0.6)	T = −5.95 ***	2.8 (0.6)	T = −5.74 ***
	Yes	2.5(0.7)		2.6 (0.7)		2.6(0.7)		2.6 (0.7)		2.6 (0.7)	
Chronic disease	None	2.6(0.7)	F = 27.93 ***	2.7(0.6)	F = 47.49 ***	2.7(0.6)	F = 76.61 ***	2.8(0.6)	F = 93.12 ***	2.9(0.6)	F = 124.49 ***
	1	2.3(0.7)		2.3(0.7)		2.3(0.6)		2.3(0.7)		2.4(0.7)	
	2	2.2(0.7)		2.2(0.7)		2.2(0.7)		2.2(0.7)		2.3(0.7)	
	Over 3	1.8(0.8)		2.0(0.8)		1.9(0.6)		1.9(0.6)		1.8(0.6)	
Smoking		2.5(0.7)	T = −1.41	2.6(0.7)	T = −2.76 **	2.6(0.7)	T = −3.46 **	2.6 (0.7)	T = −3.10 **	2.6 (0.7)	T = −3.68 ***
		2.6(0.7)		2.7(0.6)		2.7(0.7)		2.7 (0.6)		2.7 (0.6)	
Drinking		2.4(0.7)	T = 4.91 ***	2.5(0.7)	T = −7.14 ***	2.4(0.7)	T = −8.24 ***	2.5 (0.7)	T = −7.70 ***	2.4 (0.7)	T = −9.57 ***
		2.6(0.7)		2.7(0.6)		2.7(0.6)		2.7 (0.6)		2.8 (0.6)	

*p* * < 0.05, *p* ** < 0.01, *p* *** < 0.001.

**Table 3 healthcare-09-00470-t003:** A longitudinal association analysis between precarious employment and overall health status.

Variables	Model 1		Model 2		Model 3		Model 4	
	B (s.e.)	95% C.I.	B (s.e.)	95% C.I.	B (s.e.)	95% C.I.	B (s.e.)	95% C.I.
Employment status	−0.379 (0.017)	0.347–0.412	−0.364 (0.017)	0.332–0.397	−0.331 (0.016)	0.304–0.369	−0.321 (0.016)	0.294–0.359
Socioeconomic status								
Sex			0.054 (0.030)	−0.007–0.109	−0.008 (0.029)	−0.067–0.047	0.040 (0.030)	−0.020–0.096
Age			−0.061 (0.010)	−0.081–0.042	−0.052 (0.010)	−0.073–−0.034	−0.027 (0.010)	−0.047–0.009
Education			0.120 (0.012)	0.100–0.149	0.111 (0.011)	0.093–0.141	0.101 (0.011)	0.083–0.130
Injury-related variables								
Working period					0.015 (0.003)	0.071–0.157	0.014 (0.002)	0.072–0.155
Cause of injury					−0.194 (0.003)	−0.256–0.105	−0.176 (0.038)	−0.236–0.090
Claim duration					−0.106 (0.009)	−0.201–0.137	−0.099 (0.009)	−0.190–0.128
Health status								
Disability rating					0.067 (0.030)	0.053–0.165	0.075 (0.028)	0.060–0.168
Chronic disease							−0.131 (0.010)	−0.157–0.115
Smoking							−0.001 (0.015)	−0.031–0.028
Drinking							0.087 (0.015)	0.061–0.120

## Data Availability

Data are available upon reasonable request to the corresponding author.

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
