# Peer review of "The Effect of Changes in Employment on Health of Work-Related Injured Workers: A Longitudinal Perspectives"

_healthcare, 2021, doi:10.3390/healthcare9040470_

Round 1
Reviewer 1 Report
The Statistical Analysis section has many gaps.
There is no indication of the tests used (t-test, F-test).
Which p-values are considered significant? (p <0.05?).
The mention of the software used is missing.
There is no indication of the distribution of the data (if the data were not normal, non-parametric tests had to be used)
What algorithm was used to match the propensity score?
Table 2 is very confusing, and the reference of the asterisks associated with the tests is missing. Please improve its formatting and add the legend of the asterisks.
The Statistical Analysis section has many gaps.
There is no indication of the tests used (t-test, F-test).
Which p-values are considered significant? (p <0.05?).
The mention of the software used is missing.
There is no indication of the distribution of the data (if the data were not normal, non-parametric tests had to be used)
What algorithm was used to match the propensity score?
Table 2 is very confusing, and the reference of the asterisks associated with the tests is missing. Please improve its formatting and add the legend of the asterisks.
The Statistical Analysis section has many gaps.
There is no indication of the tests used (t-test, F-test).
Which p-values are considered significant? (p <0.05?).
The mention of the software used is missing.
There is no indication of the distribution of the data (if the data were not normal, non-parametric tests had to be used)
What algorithm was used to match the propensity score?
Table 2 is very confusing, and the reference of the asterisks associated with the tests is missing. Please improve its formatting and add the legend of the asterisks.
The Statistical Analysis section has many gaps.
There is no indication of the tests used (t-test, F-test).
Which p-values are considered significant? (p <0.05?).
The mention of the software used is missing.
There is no indication of the distribution of the data (if the data were not normal, non-parametric tests had to be used)
What algorithm was used to match the propensity score?
Table 2 is very confusing, and the reference of the asterisks associated with the tests is missing. Please improve its formatting and add the legend of the asterisks.
The Statistical Analysis section has many gaps.
There is no indication of the tests used (t-test, F-test).
Which p-values are considered significant? (p <0.05?).
The mention of the software used is missing.
There is no indication of the distribution of the data (if the data were not normal, non-parametric tests had to be used)
What algorithm was used to match the propensity score?
Table 2 is very confusing, and the reference of the asterisks associated with the tests is missing. Please improve its formatting and add the legend of the asterisks.
The Statistical Analysis section has many gaps.
There is no indication of the tests used (t-test, F-test).
Which p-values are considered significant? (p <0.05?).
The mention of the software used is missing.
There is no indication of the distribution of the data (if the data were not normal, non-parametric tests had to be used)
What algorithm was used to match the propensity score?
Table 2 is very confusing, and the reference of the asterisks associated with the tests is missing. Please improve its formatting and add the legend of the asterisks.
The Statistical Analysis section has many gaps.
There is no indication of the tests used (t-test, F-test).
Which p-values are considered significant? (p <0.05?).
The mention of the software used is missing.
There is no indication of the distribution of the data (if the data were not normal, non-parametric tests had to be used)
What algorithm was used to match the propensity score?
Table 2 is very confusing, and the reference of the asterisks associated with the tests is missing. Please improve its formatting and add the legend of the asterisks.
Author Response
Point 1: The Statistical Analysis section has many gaps.
Response 1: Thank you for your comments. I have checked it again and modified it briefly.
Point 2: There is no indication of the tests used (t-test, F-test).
Response 2: Thank you for your comments. In the Statistical Analysis section, you have modified the contents as follows: Next, T-test, One-way ANOVA was identified as the characteristics of the overall health status according to the characteristics of the research subjects.
Point 3: Which p-values are considered significant? (p <0.05?).
Response 3: Thank you for your comments. The significance level of this study was taken into account 0.05.
Point 4: The mention of the software used is missing.
Response 4: Thank you for your comments. In the Statistical Analysis section, you have added the contents as follows: the study used the STATA 13.0 program to perform the following analysis
Point 5: There is no indication of the distribution of the data (if the data were not normal, non-parametric tests had to be used)
Response 5: Thank you for your comments. In the Statistical Analysis section, you have added the contents as follows: We confirmed that the absolute value of skewness is less than 3 and the kurtosis is less than 10 and that the normality assumption is satisfied
Point 6: What algorithm was used to match the propensity score?
Response 6: Thank you for your comments. In the Statistical Analysis section, you have added the contents as follows: The study used Nearest Neighbor Matching with a Caliper (1:1), a matching method that can reduce the occurrence of convenience, focusing on minimizing heterogeneity between groups according to employment status.
Point 7: Table 2 is very confusing, and the reference of the asterisks associated with the tests is missing. Please improve its formatting and add the legend of the asterisks.
Response 7: Thank you for your comments. Table 2 was modified briefly and the significance level range was displayed.
Reviewer 2 Report
Abstract
“This longitudinal study attempts to identify changes in employment status and overall health status, after employment, of workers who experienced industrial accidents.” Evaluate experienced industrial accidents or in current work? Please explain better and specify that the study was carried out on Korean workers
“The research data used was from the primary cohort (2013-2017) of the Panel Study of Workers’ Compensation Insurance” Why primary? are there others?
“A, 1,853 were selected for the study from, excluding those with missing values.” Please write better
“chaotic variables” Please explain better
“The significance of the study can be found in that PSM was used to clarify the causal relationship” Please explain better
Introduction
“The factors related to labor include employment type, wage level, and working environment; these factors are reported to affect the health of individuals directly or indirectly” Please insert the ref.
“Even after considering the expansion of industrial accident insurance policies, the number is high and increasing steadily each year.” Where? In Korea or in all world?
- Materials and Methods
2.1. Data Source and Study Population
“the first cohort (2013- 2017) of the PSWCI” Please explain why you have done the research over these years
“A total of 1,853 people participated in 2012, divided into 1,268 employees and 535 unemployed workers by employment status.” what are they referring to? not part of the sample, right?
Table 1. General characteristics of workers with industrial accident experience
Please explain under years what are the numbers (% or average or mediane cc)
Discussion
“This study is a longitudinal study to identify the relationship between the employment status and overall health status of workers experiencing industrial accidents “ Please that the study is on korean workers.
“chaotic variables” Please explain better
“The association between employment status and overall health status is consistent with the results of previous studies” Please insert the ref.
“Studies on employment status, health, and quality of life of people with multiple neurological disorders shows that those with certain disabilities have more experience with job loss, which results in deterioration of health status and deterioration in quality of life “ Please see you
Fabiola Silvaggi, Matilde Leonardi, Pietro Tiraboschi, Cristina Muscio, Claudia Toppo, Alberto Raggi, Keeping People with Dementia or Mild Cognitive Impairment in Employment: A Literature Review on Its Determinants, Int. J. Environ. Res. Public Health 2020, 17(3), 842. doi:10.3390/ijerph17030842
Silvaggi Fabiola, Leonardi Matilde, Raggi Alberto, Eigenmann Michela, Mariniello Arianna, Silvani
Antonio, Lamperti Elena, Schiavolin Silvia, Employment and Work Ability of Persons With Brain Tumors: A Systematic Review, Frontiers in Human Neuroscience, 2020, 14,452. doi: 10.3389/fnhum.2020.571191
“Secondly……” it does not detect any news not yet discovered
“Thirdly….” It is this the main result of study and it should be commented in more detail
Conclusion
“Meanwhile, this study has some lim-itations.” I suggest to move this paragraph into discussion
The conclusion must be written. What is there does not reflect any conclusions
Author Response
2-1. Abstract
Point 1: This longitudinal study attempts to identify changes in employment status and overall health status, after employment, of workers who experienced industrial accidents.” Evaluate experienced industrial accidents or in current work? Please explain better and specify that the study was carried out on Korean workers
Response 1: Thank you for your valuable comment. This longitudinal study was conducted on people who have experienced work-related injuries in the past. In Korea, there are many social programs to help return to work for people who experienced work-related injuries. Also, we have added this in the abstract section [page 1, lines 17-18]
Point 2: “The research data used was from the primary cohort (2013-2017) of the Panel Study of Workers’ Compensation Insurance” Why primary? are there others?
Response 2: Thank you for your feedback. We have elaborated on this aspect of our study in the relevant sub-explanation of dataset.
[page 1, line 18] In this study, we used the Panel Study of Workers’ Compensation Insurance from 2013 to 2017.
Point 3: “A, 1,853 were selected for the study from, excluding those with missing values.” Please write better
Response 3: Thank you for your bringing this aspect of our methods to our attention, as it needs additional explanation. We have revised the abstract and methods section to read clearly. Instead of deleting “A, 1,853 were selected for the study from, excluding those with missing values” from the abstract section, we have added a detailed explanation for population in the methods [page 2, lines 87-91].
Point 4: “chaotic variables” Please explain better
Response 4: Thank you for your comments. In line with your comments, we have replaced the term “chaotic variables” with “confounding variables” [page 1, line 20].
Point 5: “The significance of the study can be found in that PSM was used to clarify the causal relationship” Please explain better
Response 5: Thank you for bringing to our attention the insufficient information provided in the abstract section. Accordingly, we have revised the abstract section of the manuscript.
[page 1, lines 26-27] The notable point of this study was to apply the PSM methods. By applying PSM, we clearly identified the effect of changes in employment status on health status.
- Introduction
Point 6: “The factors related to labor include employment type, wage level, and working environment; these factors are reported to affect the health of individuals directly or indirectly” Please insert the ref.
Response 6: Thank you for your feedback of our insufficient part of introduction. We have added the reference for the sentence [page 1, lines 35-36]
- Rönnblad, T., Grönholm, E., Jonsson, J., Koranyi, I., Orellana, C., Kreshpaj, B., . . . Bodin, T. (2019). Precarious employment and mental health: a systematic review and meta-analysis of longitudinal studies. Scand J Work Environ Health, 45(5), 429-443. doi:10.5271/sjweh.3797
- Virtanen, P., Vahtera, J., Kivimäki, M., Pentti, J., & Ferrie, J. (2002). Employment security and health. J Epidemiol Community Health, 56(8), 569-574. doi:10.1136/jech.56.8.569
- van der Velden, P. G., Muffels, R. J. A., Peijen, R., & Bosmans, M. W. G. (2019). Wages and employment security following a major disaster: A 17-year population-based longitudinal comparative study. PLoS One, 14(3), e0214208. doi:10.1371/journal.pone.0214208
Point 7: “Even after considering the expansion of industrial accident insurance policies, the number is high and increasing steadily each year.” Where? In Korea or in all world?
Response 7: Thank you for your comments. We have revised the introduction section of the manuscript to read clearly.
[page 2, lines 50-52] Even after considering the expansion of industrial accident insurance policies, the number of work-related injuries is high and increasing steadily each year in Korea.
- Materials and Methods
- Data Source and Study Population
Point 8: “the first cohort (2013- 2017) of the PSWCI” Please explain why you have done the research over these years
Response 8: Thank you for bringing to our attention the insufficient information provided in the Materials and Methods section. Accordingly, we have provided the required details of data.
[page 2, lines 79-82] This study was a longitudinal study using the first to fifth years of Panel Study of Workers’ Compensation Insurance (PSWCI) data on workers experienced work-related injuries. Because of the purpose in this study, identifying association between changes in employment status and health, we have to use the longitudinal data less than 3-year.
Point 9: “A total of 1,853 people participated in 2012, divided into 1,268 employees and 535 unemployed workers by employment status.” what are they referring to? not part of the sample, right?
Response 9: Thank you for your query. Our description in the section of data source and study population was misleading. So, we have revised and added the explanation of our study population follow:
[page 2, lines 87-91] The 1,958 participants in 2012 were workers who experienced work-related injuries before applying propensity score matching. In 2012, the participants were consisted with 1,387 employed workers and 571 unemployed workers. After we applied the PSM method, the final participants in this study were 571 employed workers and 571 unemployed workers.
Point 10: Table 1. General characteristics of workers with industrial accident experience Please explain under years what are the numbers (% or average or mediane cc)
Response 10: Thank you for your feedback. We have added N or % in the Table 1. Also, there is no continuous variables in Table 1.
- Discussion
Point 11: “This study is a longitudinal study to identify the relationship between the employment status and overall health status of workers experiencing industrial accidents “ Please that the study is on korean workers.
Response 11: Thank you for bringing this in accuracy to our attention. We have revised the relevant sentence of the manuscript to confirm the association between changes in employment status and health in Korean worker who experienced work-related injuries.
[page 6, lines 199-201] This study is a longitudinal study to identify the association between the employment status and overall health status of workers who experienced work-related injuries in Korea.
Point 12: “chaotic variables” Please explain better
Response 12: Thank you for your feedback. We have already replaced the term “chaotic variables” with “confounding variables” in Abstract and Discussion section [page 6, line 203].
Point 13: “The association between employment status and overall health status is consistent with the results of previous studies” Please insert the ref.
Response 13: Thank you for your comments. Per your recommendation, we have added the relevant references [page 6, lines 212-213].
- Marck, C. H., Aitken, Z., Simpson, S., Jr., Weiland, T. J., Kavanagh, A., & Jelinek, G. A. (2020). Predictors of Change in Employment Status and Associations with Quality of Life: A Prospective International Study of People with Multiple Sclerosis. J Occup Rehabil, 30(1), 105-114. doi:10.1007/s10926-019-09850-5
- Choi, J. W., Kim, J., Han, E., & Kim, T. H. (2019). Association of employment status and income with self-rated health among waged workers with disabilities in South Korea: population-based panel study. BMJ Open, 9(11), e032174. doi:10.1136/bmjopen-2019-032174
- Park, S. J., Kim, S. Y., Lee, E. S., & Park, S. (2020). Associations among Employment Status, Health Behaviors, and Mental Health in a Representative Sample of South Koreans. Int J Environ Res Public Health, 17(7). doi:10.3390/ijerph17072456
Point 14: “Studies on employment status, health, and quality of life of people with multiple neurological disorders shows that those with certain disabilities have more experience with job loss, which results in deterioration of health status and deterioration in quality of life “ Please see you
Fabiola Silvaggi, Matilde Leonardi, Pietro Tiraboschi, Cristina Muscio, Claudia Toppo, Alberto Raggi, Keeping People with Dementia or Mild Cognitive Impairment in Employment: A Literature Review on Its Determinants, Int. J. Environ. Res. Public Health 2020, 17(3), 842. doi:10.3390/ijerph17030842
Silvaggi Fabiola, Leonardi Matilde, Raggi Alberto, Eigenmann Michela, Mariniello Arianna, Silvani
Antonio, Lamperti Elena, Schiavolin Silvia, Employment and Work Ability of Persons With Brain Tumors: A Systematic Review, Frontiers in Human Neuroscience, 2020, 14,452. doi: 10.3389/fnhum.2020.571191
Response 14: Thank you for your comments. We have read the references you introduced in detail and reflected it in our manuscripts.
[page 6, lines 217-220] However, some studies on the employment of people with neurological disorders argued that many people with disabilities were still employed, and additional political supports such as vocational rehabilitation were needed to maintain and improve their employment.
Point 15: “Secondly……” it does not detect any news not yet discovered
Response 15: Thank you for your comment. The subgroup analysis was conducted for dealing with this. Also, we added our supplementary.
Point 16: “Thirdly….” It is this the main result of study and it should be commented in more detail
Response 16: Thank you for your valuable comment. As your suggestion, we have revised and added the discussion section of the manuscript.
[page 7, lines 256-259] Based on the results of the first year’s analysis of this study, 1,792 workers (91.5%) suffered industrial accidents due to accidents, and 166 workers (8.5%) suffered from diseases (Table 1). Also, even after PSM was applied, the disease among the causes of industrial accident has a negative effect on health in Korea.
[pages 7, lines 265-270] Many studies about systematic risk underestimation of worker’s risk suggested that people make systematic errors in their perception and predictions as current and past emotions influence assessments. In other word, the frequency of dramatic or sensational events such as causes of death were mor overestimated, and the frequency of less well-publicized causes such as stroke, asthma was more underestimated.
- Conclusion
Point 17: “Meanwhile, this study has some lim-itations.” I suggest to move this paragraph into discussion
Response 17: Thank you for your feedback. Per your recommendation, we have revised sentence “Meanwhile, …” by moving from the conclusion section to the discussion section [pages 7-8, lines 276-283].
Point 18: The conclusion must be written. What is there does not reflect any conclusions
Response 18: We are really thanked you for your valuable comment. As your recommendation, we have developed conclusion section of the manuscript. Results of the study will pay attention to additional support to workers who experienced work-related injuries. Because changes in employment (employment/unemployment) have impacted on their health.
[pages 8, liens 287-289] The results of the study suggest that the longitudinal and consecutive political support and subsidy will be required to workers who experienced work-related injuries, even they returned to the work.

Reviewer 3 Report
Ref: healthcare-1139085
Journal: Healthcare (ISSN 2227-9032), MDPI
Title: The Longitudinal Association Between the Employment Status and Health of Work-related Injured Workers: Applying Propensity Score Matching
- General comments
- This study is interesting and attempts to identify changes in employment status and overall health status, after employment, of workers who experienced industrial accidents.
- The topic of this manuscript is appropriate for Healthcare journal
- The overall presentation of the paper is good
- The paper’s length is suitable for contribution
- The literature list includes over 20 items.
- Nevertheless, this is not to say that it is without blemish. I have several comments regarding the scientific part of the manuscript and I expect the authors to proceed in a resubmission after a reformation of their article.
- In the current version of the manuscript, the authors did not explain clearly what the novelty and key points of this study are.
- Moreover, taking into consideration that there are several limitations in this study due to the assumptions made in the statistical analysis of the data, a suggestion for further research, and also some proposed scientific directions, have to be incorporated in the last paragraphs.
- A nomenclature incorporated in the end of the manuscript, would be very useful.
- The manuscript’s similarity-index (by using the TURNITIN checker) is low, i.e. it is equal to 19%.
- Specific comments
Title:
- The paper’s title could be reformed in order to be more accurate as far as the paper’s aim is concerned.
Abstract:
- The abstract could be reformed in order to present in a clearer way, the deductions of the work.
Discussion:
- This section is quite cursory, and could be revised. The authors did not discuss clearly on what the insights are from this new study and how these insights can be generalized to other situations.
- In the current version of the manuscript, the authors did not explain clearly what the novelty and key points of this study are.
- Please, rewrite the discussion and the conclusions in order to respond to the aims and prove the hypothesis of the study, in a clear way.
Decision: My opinion is the manuscript must be resubmitted, after a minor revision of it.
Attached document: Please see the attached documents with reviewer’s comments.
Author Response
Point 1: Title: The paper’s title could be reformed in order to be more accurate as far as the paper’s aim is concerned.
Response 1: Thank you for your valuable comment. We have changed title of the manuscript to be more accurate.
[page 1, liens 2-3] The Effect of Changes in Employment on Health of Work-related Injured Workers: A Longitudinal Perspectives”
Abstract:
Point 2: The abstract could be reformed in order to present in a clearer way, the deductions of the work.
Response 2: Thank you for your comments. We have revised our abstract section of the manuscript more detail [page 1, liens 16-27].
Discussion:
Point 3: This section is quite cursory, and could be revised.
Response 3: Thank you for your comment. In response to your opinion, we have drastically revised discussion section of the manuscript.
[page 6, liens 199-201] This study is a longitudinal study to identify the association between the employ-ment status and overall health status of workers who experienced work-related injuries in Korea.
[page 6, liens 212-213] The association between employment status and overall health status is consistent with the results of previous studies.
[page 6, liens 217-219] However, some studies on the employment of people with neurological disorders argued that many people with disabilities were still employed, and additional political supports such as vocational rehabilitation were needed to maintain and improve their employment.
[page 7, liens 256-259] Based on the results of the first year’s analysis of this study, 1,792 workers (91.5%) suffered industrial accidents due to accidents, and 166 workers (8.5%) suffered from diseases (Table 1). Also, even after PSM was applied, the disease among the causes of industrial accident has a negative effect on health in Korea.
[page 7, liens 265-270] Many studies about systematic risk underestimation of worker’s risk suggested that people make systematic errors in their perception and predictions as current and past emotions influence assessments. In other word, the frequency of dramatic or sensational events such as causes of death were mor overestimated, and the frequency of less well-publicized causes such as stroke, asthma was more underestimated.
[pages 7-8, liens 276-283] Meanwhile, this study has some limitations. Because secondary data (the panel study of workers' compensation insurance) was used, it was not possible to analyze the factors influencing workers' health status extensively. In other words, there were limits to the available variables. For example, there was a lack of variables that could reflect material status (e.g., household income), family, and social networks. The employment group was divided into restricted groups, and it was limited to reflecting all employment instability. In addition, due to the sporadic nature of the data, it was not suitable to confirm the continuous impact of employment instability.
Point 4: The authors did not discuss clearly on what the insights are from this new study and how these insights can be generalized to other situations.
Response 4: Thank you for bringing this in accuracy to our attention. This study was used data from PSWCI in Korea. Despite of local setting, this population-based study will be generalized to work-related injured workers.
Point 5: In the current version of the manuscript, the authors did not explain clearly what the novelty and key points of this study are.
Please, rewrite the discussion and the conclusions in order to respond to the aims and prove the hypothesis of the study, in a clear way.
Response 5: Thank you for bringing our attention. Per your suggestion, we have revised and added discussion and conclusion section of the manuscript.
